## [Reviewer comments · Royal Society Open Science]

Review History

RSOS-191255.R0 (Original submission)

Review form: Reviewer 1

Is the manuscript scientifically sound in its present form?

Yes

Are the interpretations and conclusions justified by the results?

Yes

Is the language acceptable?

Yes

Do you have any ethical concerns with this paper?

No

Have you any concerns about statistical analyses in this paper?

No

Recommendation?

Accept with minor revision (please list in comments)

Comments to the Author(s)

I'm quite impressed. This is an extremely clear paper introducing and studying (using numerical simulation) a framework to model the interplay between imitation and serendipity in situations where agents are ranked (and aware of being ranked). The results are intriguing and, at times, provocative. I also appreciate the fact that the code is publicly available. I only have a small set of minor comments.

Abstract

The abstract does not report the method being used. Is it a numerical simulation, or a theoretical/mathematical analysis, or an empirical experiment? (At this point of the reading, this is not clear at all).

Introduction

Very well written. No particular comments.

Model

Very clear. No particular comments.

Results

Utility and inequality

Extremely well written. No particular comments.

Meritocracy and homogenisation

I think that the "meritocracy part" of this section can be improved. What are the Kendall correlation? What do they represent? Please explain.

Social mobility

Please define Heaviside's step function.

In Eq 6, you define Δm_i , but then in Figure 5 you plot Δm . What is the relationship between Δm_i and Δm ? Please explain.

Discussion

Please add a paragraph regarding future work.

Review form: Reviewer 2

Is the manuscript scientifically sound in its present form?

Yes

Are the interpretations and conclusions justified by the results?

Yes

Is the language acceptable?

Yes

Do you have any ethical concerns with this paper?

No

Have you any concerns about statistical analyses in this paper?

No

Recommendation?

Accept with minor revision (please list in comments)

Comments to the Author(s)

This paper presents an agent-based model of imitation and innovation when agents try to compete for the ranks. The author shows that high imitation results in the inability of agents to find their optimal utility, and it further induces the ranking inequality. In general, the paper is well-written, and despite the simplicity of the model, it provides an interesting insight. However, there are a couple of unclear points in the paper that need to be addressed by the author. Besides, the paper lacks more discussion on the limitation of the model and the assumptions. My comments are detailed below:

- The major limitation of the model is that it lacks the social processes behind the adoption and evaluation of actions. This needs to be acknowledged and explained by the author. Additionally, the payoff can increase as some actions reach higher ranks.
- Referring to the previous point, the model lacks to provide insights on the impact of the network structure on ranking. For example, in a recent paper by Karimi et al. Sci. Rep (2018), the authors showed that the structure itself could generate ranking inequalities, and with the addition of imitation, this initial inequality can be reinforced.
- Line 56: serendipity should be defined and explained.
- Figure 1: In the illustration of the model it would be helpful to guide the reader on the three components of the model by adding an explanation next to each array.
- Section B: "fitness" is a particular concept in these types of agent-based models and models of network formation, and it refers typically to an intrinsic property of an agent. However, in this model, it indicates a different strategy of the agents to choose actions. Therefore I suggest the author choose a better terminology here, e.g. "strategy", that would fit the nature of the action better.
- In section B, it is not clear whether all the agents play a diversity strategy or maximizing strategy. The author should clarify that.
- Results in Fig. 4 require more clarification. Firstly, it is puzzling that when agents play the maximizing strategy, their utility is not completely correlated with the strategy. I would guess that according to eq. 1, the individual's utility should have a correlation close to 1 with maximizing strategy when $q = 0$.
- In Figure 4, how would the author interpret the tipping point at $q = 0.2$? Are these results achieved once the agents have stabilized their ranking position?
- The author should provide information about the robustness of the results. Are the results dependent on the choice of the initial parameters?
- In section B if agents always play a diversifying or maximizing strategy, it is not clear how it would be related to q . The author should clarify that.

- Section C: the term “Social mobility” is misleading here. Social would indicate that agents make a decision based on a social aspect, and mobility would mean that agents mobilize themselves. However, the results that are presented in Fig. 5 are related to the evolution or dynamic of the ranking positions.

Decision letter (RSOS-191255.R0)

16-Sep-2019

Dear Dr Livan

On behalf of the Editors, I am pleased to inform you that your Manuscript RSOS-191255 entitled "Don't follow the leader: How ranking performance reduces meritocracy" has been accepted for publication in Royal Society Open Science subject to minor revision in accordance with the referee suggestions. Please find the referees' comments at the end of this email.

The reviewers and handling editors have recommended publication, but also suggest some minor revisions to your manuscript. Therefore, I invite you to respond to the comments and revise your manuscript.

- Ethics statement

- Data accessibility

<http://datadryad.org/submit?journalID=RSOS&manu=RSOS-191255>

- Competing interests

- Authors' contributions

All submissions, other than those with a single author, must include an Authors' Contributions section which individually lists the specific contribution of each author. The list of Authors should meet all of the following criteria; 1) substantial contributions to conception and design, or

acquisition of data, or analysis and interpretation of data; 2) drafting the article or revising it critically for important intellectual content; and 3) final approval of the version to be published.

- Acknowledgements

- Funding statement

Because the schedule for publication is very tight, it is a condition of publication that you submit the revised version of your manuscript before 25-Sep-2019. Please note that the revision deadline will expire at 00.00am on this date. If you do not think you will be able to meet this date please let me know immediately.

- 1) A text file of the manuscript (tex, txt, rtf, docx or doc), references, tables (including captions) and figure captions. Do not upload a PDF as your "Main Document";

- 2) A separate electronic file of each figure (EPS or print-quality PDF preferred (either format should be produced directly from original creation package), or original software format);
- 3) Included a 100 word media summary of your paper when requested at submission. Please ensure you have entered correct contact details (email, institution and telephone) in your user account;
- 4) Included the raw data to support the claims made in your paper. You can either include your data as electronic supplementary material or upload to a repository and include the relevant doi within your manuscript. Make sure it is clear in your data accessibility statement how the data can be accessed;
- 5) All supplementary materials accompanying an accepted article will be treated as in their final form. Note that the Royal Society will neither edit nor typeset supplementary material and it will be hosted as provided. Please ensure that the supplementary material includes the paper details where possible (authors, article title, journal name).

If your manuscript is newly submitted and subsequently accepted for publication, you will be asked to pay the article processing charge, unless you request a waiver and this is approved by Royal Society Publishing. You can find out more about the charges at <http://rsos.royalsocietypublishing.org/page/charges>. Should you have any queries, please contact openseience@royalsociety.org.

Kind regards,
Andrew Dunn
Royal Society Open Science Editorial Office
Royal Society Open Science
openseience@royalsociety.org

on behalf of Professor Matjaz Perc (Associate Editor) and Marta Kwiatkowska (Subject Editor)
openseience@royalsociety.org

Reviewer comments to Author:
Reviewer: 1

Comments to the Author(s)

I'm quite impressed. This is an extremely clear paper introducing and studying (using numerical simulation) a framework to model the interplay between imitation and serendipity in situations

where agents are ranked (and aware of being ranked). The results are intriguing and, at times, provocative. I also appreciate the fact that the code is publicly available. I only have a small set of minor comments.

Abstract

The abstract does not report the method being used. Is it a numerical simulation, or a theoretical/mathematical analysis, or an empirical experiment? (At this point of the reading, this is not clear at all).

Introduction

Very well written. No particular comments.

Model

Very clear. No particular comments.

Results

Utility and inequality

Extremely well written. No particular comments.

Meritocracy and homogenisation

I think that the “meritocracy part” of this section can be improved. What are the Kendall correlation? What do they represent? Please explain.

Social mobility

Please define Heaviside’s step function.

In Eq 6, you define Δm_i , but then in Figure 5 you plot Δm . What is the relationship between Δm_i and Δm ? Please explain.

Discussion

Please add a paragraph regarding future work.

Reviewer: 2

Comments to the Author(s)

This paper presents an agent-based model of imitation and innovation when agents try to compete for the ranks. The author shows that high imitation results in the inability of agents to find their optimal utility, and it further induces the ranking inequality. In general, the paper is well-written, and despite the simplicity of the model, it provides an interesting insight. However, there are a couple of unclear points in the paper that need to be addressed by the author. Besides, the paper lacks more discussion on the limitation of the model and the assumptions. My comments are detailed below:

- The major limitation of the model is that it lacks the social processes behind the adoption and evaluation of actions. This needs to be acknowledged and explained by the author. Additionally, the payoff can increase as some actions reach higher ranks.
- Referring to the previous point, the model lacks to provide insights on the impact of the network structure on ranking. For example, in a recent paper by Karimi et al. Sci. Rep (2018), the authors showed that the structure itself could generate ranking inequalities, and with the addition of imitation, this initial inequality can be reinforced.
- Line 56: serendipity should be defined and explained.
- Figure 1: In the illustration of the model it would be helpful to guide the reader on the three components of the model by adding an explanation next to each array.
- Section B: "fitness" is a particular concept in these types of agent-based models and models of network formation, and it refers typically to an intrinsic property of an agent. However, in this model, it indicates a different strategy of the agents to choose actions. Therefore I suggest the author choose a better terminology here, e.g. "strategy", that would fit the nature of the action better.
- In section B, it is not clear whether all the agents play a diversity strategy or maximizing strategy. The author should clarify that.
- Results in Fig. 4 require more clarification. Firstly, it is puzzling that when agents play the maximizing strategy, their utility is not completely correlated with the strategy. I would guess that according to eq. 1, the individual's utility should have a correlation close to 1 with maximizing strategy when $q = 0$.
- In Figure 4, how would the author interpret the tipping point at $q = 0.2$? Are these results achieved once the agents have stabilized their ranking position?
- The author should provide information about the robustness of the results. Are the results dependent on the choice of the initial parameters?
- In section B if agents always play a diversifying or maximizing strategy, it is not clear how it would be related to q . The author should clarify that.
- Section C: the term "Social mobility" is misleading here. Social would indicate that agents make a decision based on a social aspect, and mobility would mean that agents mobilize themselves. However, the results that are presented in Fig. 5 are related to the evolution or dynamic of the ranking positions.

Author's Response to Decision Letter for (RSOS-191255.R0)

See Appendices A & B.

Decision letter (RSOS-191255.R1)

24-Sep-2019

Dear Dr Livan,

I am pleased to inform you that your manuscript entitled "Don't follow the leader: How ranking performance reduces meritocracy" is now accepted for publication in Royal Society Open Science.

on behalf of Professor Matjaz Perc (Associate Editor) and Marta Kwiatkowska (Subject Editor)
openscience@royalsociety.org

Appendix A

Response to the reviews of submission RSOS-191255

Don't follow the leader: How ranking performance reduces meritocracy

I would like to thank both Reviewers for their comments and feedback, which gave me the opportunity to substantially improve the paper. I have made several changes following their suggestions, particularly in the final Discussion section. In the following, I reply to all comments from the Reviewers in the same order as they appear in their reports.

Reviewer #1

The abstract does not report the method being used. Is it a numerical simulation, or a theoretical/mathematical analysis, or an empirical experiment? (At this point of the reading, this is not clear at all).

I modified the abstract in order to specify that the research question I tackle is addressed by “numerically simulating” an agent-based model.

*Meritocracy and homogenisation:
I think that the “meritocracy part” of this section can be improved. What are the Kendall correlation? What do they represent? Please explain.*

I have added the definition of Kendall's correlation coefficient and a reference (Ref. [32]) to the original paper that introduced it.

*Social mobility:
Please define Heaviside's step function.
In Eq. 6, you define Δm_i , but then in Figure 5 you plot Δm . What is the relationship between Δm_i and Δm ? Please explain.*

I have added the definition of Heaviside's step function. I thank the Reviewer for spotting the inconsistency between the text and the labels in Figure 5. I have changed the latter to m_i and Δm_i on the x and y axes,

respectively. Also, it should be noted that the title of this section is no longer “social mobility”: following the recommendations of Reviewer #2 I have changed the title to “ranking dynamics”.

Please add a paragraph regarding future work.

I have substantially changed the final part of the Discussion in order to mention future work. I mention two main areas of possible future development. First, I discuss possible ways to make contact between the model’s predictions and real-world data, mentioning academic citation data as the ideal “laboratory” where to test the relationship between imitation, serendipity, and meritocracy. Second, I discuss possible extensions of the model in order to make it less stylised. In particular, I mention possible ways to make the agents’ preference towards serendipity or imitation adaptive and responsive to past outcomes. In addition, I also discuss the fact that the model’s dynamics, in its current form, does not lead to qualitatively different results when run in a well mixed population (as in the paper) or on a network. This is in partial contrast with previously published literature (see, e.g., Ref. [39]), which shows that network topology alone can determine ranking outcomes, and future extensions of the model should be able to account for that.

Reviewer #2

The major limitation of the model is that it lacks the social processes behind the adoption and evaluation of actions. This needs to be acknowledged and explained by the author. Additionally, the payoff can increase as some actions reach higher ranks.

I absolutely agree with this point, which was not stated clearly enough in the previous submission. I have added a paragraph at the end of the “the model” section in order to highlight the main limitations of the model, and to reiterate that the model is only a stylised representation of much more complex real-life dynamics.

Referring to the previous point, the model lacks to provide insights on the impact of the network structure on ranking. For example, in a recent paper by Karimi et al. Sci. Rep (2018), the authors showed that the structure itself could generate ranking inequalities, and with the addition of imitation, this initial inequality can be reinforced.

I thank the Reviewer for pointing out this paper, which I was not aware of. I added a reference to it (Ref. [39]) and used it as a benchmark to discuss possible extensions of the model on a network. Indeed, the model presented in the paper does not lead to any major change in results if implemented on a network, i.e., by constraining the agents to only imitate the actions adopted by their more successful neighbours. This is because any connected network topology would ultimately allow the imitation process of the model to diffuse, therefore leading to results qualitatively identical to those obtained with the well mixed population approach adopted in the paper. I have clarified this point in the final Discussion section, when mentioning possible future extensions of the model.

Line 56: serendipity should be defined and explained.

I have added a definition of serendipity where suggested by the Reviewer.

Figure 1: In the illustration of the model it would be helpful to guide the reader on the three components of the model by adding an explanation next to each array.

I thank the Reviewer for making this suggestion, I have amended the figure accordingly.

Section B: “fitness” is a particular concept in these types of agent-based models and models of network formation, and it refers typically to an intrinsic property of an agent. However, in this model, it indicates a different strategy of the agents to choose actions. Therefore I suggest the author choose a better terminology here, e.g. “strategy”, that would fit the nature of the action better.

The quantities I refer to as fitnesses in the paper do measure intrinsic properties of the agents. I suppose the previous version of the paper was not clear enough in this respect, so I made changes in order to clarify this point. Both notions of fitness introduced in the paper measure the average payoff that the agents would be able to achieve if they played actions according to *predetermined* fixed strategies (i.e., not according to the model’s probabilistic dynamics). The fitness defined as ϕ^{avg} measures the average payoff the agents would receive by uniformly sampling the action space at random, while the one defined as ϕ^{max} measures the payoff the agents would receive by always playing their most profitable strategy. As mentioned in the paper, the two notions capture different aspects. The former is higher for agents who are on average good at most playing most actions, while the latter captures excellence in a single action. I hope the changes I have made to the paper are sufficient to clarify these points. Therefore, for the moment I have kept the word “fitness” to describe the above quantities. However, should the Reviewer still feel that this may be confusing I will be happy to amend the paper accordingly.

In section B, it is not clear whether all the agents play a diversity strategy or maximizing strategy. The author should clarify that. Results in Fig. 4 require more clarification. Firstly, it is puzzling that when agents play the maximizing strategy, their utility is not completely correlated with the strategy. I would guess that according to eq. 1, the individual’s utility should have a correlation close to 1 with maximizing strategy when $q = 0$.

Throughout the paper the results presented are those obtained with the same dynamics (illustrated in Fig. 1). As mentioned above, I have clarified that the quantities introduced in Eqs. (4) and (5) are intrinsic properties of the agents and do not indicate different strategies that the agents may play when selecting actions. The agents always choose actions based on the model’s probabilistic dynamics, which is why their fitnesses are only partially correlated with their utility.

The above comments by the Reviewer also made me realise that another point may be confusing, i.e., why the correlation between the fitness ϕ^{avg} and the utility is not 1 when $q = 0$. Indeed, ϕ^{avg} represents the average payoff the agents would accrue over a long time if they sampled the action space uniformly at

random. However, even when $q = 0$ the agents do not sample such space uniformly, as they play repeatedly over time the actions that yield them high payoffs, and quickly dismiss those that do not. I have added a few sentences to clarify this.

In Figure 4, how would the author interpret the tipping point at $q = 0.2$? Are these results achieved once the agents have stabilized their ranking position?

All the simulation results presented in the paper are obtained in the model's stationary state, i.e., when the ranking of individual agents may still change over time but not their expected movement across the ranking given their position in it (see, e.g., the two bottom panels in Fig. 5). Due to the lack of analytical tractability, I do not have a definitive answer to the first part of the question. However, intuitively speaking, the maximum in correlation between the fitness ϕ^{\max} and utility is due to the compromise between two competing forces. For relatively low values of q , the agents are essentially free to sample the space of available actions (i.e., those with values of P_{ij} close to 1). These will be played repeatedly over time, leading to a fast increase in utility, but will be eventually left to try out new actions (since $P_{ij} < 1, \forall i, j$). When this happens, the agents will start the sampling process again in a semi-random fashion (due to low q) until they eventually stumble again on the same highly profitable action and play it repeatedly again for some time. In contrast, high values of q will prevent most agents from performing this sampling process (which, albeit inefficiently, allows the return to profitable actions), leading them to play the actions adopted by top-ranked agents and ultimately reducing their utility. The aforementioned maximum occurs when the "compromise" between such two competing forces is optimal.

The author should provide information about the robustness of the results. Are the results dependent on the choice of the initial parameters?

From a qualitative standpoint, none of the model's results depend on the parameter specification. That is, different specifications of N and M may lead to different numerical values for the quantities of interest (e.g., utility, Gini coefficient, etc.), but the model's behaviour as a function of q remains unchanged. In the Figure below I have added a few plots to show that this is indeed the case. The first row shows results obtained while keeping $N = 100$ constant, while the second one shows results obtained for $M = 1000$ constant. The quantities I report are the Gini coefficient of the agents' utility distribution at the end of simulations (left plots), the correlation between the agents' fitness ϕ^{avg} and their utility (middle plots), and the correlation between the agents' fitness ϕ^{\max} and their utility (right plots). As it can be seen (also comparing with the middle panel of Fig. 3 and with the left panel of Fig. 4 of the main paper), changes in the parameters do not lead to qualitative changes in behaviour. In order to keep the paper light, I have added a mention of this at the end of the Results section. However, should the Reviewer feel that paper would improve substantially from the addition of a section devoted to a robustness analysis, I would be happy to add that to the paper.

In section B if agents always play a diversifying or maximizing strategy, it is not clear how it would be related to q . The author should clarify that.

As clarified above and in the paper, the utility measures I use in the paper do not refer to strategies played by the agents but rather to their intrinsic properties. As such, Fig. 4 illustrate the relationship

between ranking outcomes and such properties. In other words, this shows under what conditions on q the dynamics of the model captures well the intrinsic fitness of the agents and ranks them accordingly. In the case of ϕ^{avg} the best case scenario occurs at $q = 0$, while for ϕ^{max} it occurs at an intermediate value of q (see above reply).

Section C: the term “Social mobility” is misleading here. Social would indicate that agents make a decision based on a social aspect, and mobility would mean that agents mobilize themselves. However, the results that are presented in Fig. 5 are related to the evolution or dynamic of the ranking positions.

I have changed the wording in the paper following the Reviewer’s comment, replacing “social mobility” with “ranking dynamics” or “ranking changes”.

Additional changes

In the originally submitted manuscript Fig. 2 reported results obtained with a parameter set different from the one reported in the caption. I have changed the Figure in order to fix that.

Appendix B

Don't follow the leader: How ranking performance reduces meritocracy

Giacomo Livan^{1,2}

¹*Department of Computer Science, University College London, Gower Street, London WC1E 6EA, UK*

²*Systemic Risk Centre, London School of Economics and Political Sciences, Houghton Street, London WC2A 2AE, UK*

In the name of meritocracy, modern economies devote increasing amounts of resources to quantifying and ranking the performance of individuals and organisations. Rankings send out powerful signals, which lead to identify the actions of top performers as the ‘best practices’ that others should also adopt. However, several studies have shown that the imitation of best practices often leads to a drop in performance. So, should those lagging behind in a ranking imitate top performers or should they instead pursue a strategy of their own? I tackle this question **by numerically simulating** a stylised model of a society whose agents seek to climb a ranking either by imitating the actions of top performers or by randomly trying out different actions, i.e., via serendipity. The model gives rise to a rich phenomenology, showing that the imitation of top performers increases welfare overall, but at the cost of higher inequality. Indeed, the imitation of top performers turns out to be a self-defeating strategy that consolidates the early advantage of a few lucky - and not necessarily talented - winners, leading to a very unequal, homogenised, and effectively non-meritocratic society. Conversely, serendipity favours meritocratic outcomes and **prevents rankings from freezing**.

I. INTRODUCTION

Modern advanced economies devote ever increasing amounts of resources to quantifying and ranking the performance of individuals, companies and institutions. The rationale underpinning this trend is that of meritocracy: ranking performance encourages to strive to be at the top, generating a virtuous cycle which rewards top performers and incentivises others to improve.

Based on such rationale, both ‘white-collar’ [1, 2] and ‘blue-collar’ [3] workers in a variety of sectors are nowadays monitored and ranked based on their productivity. In many work environments, this meritocratic paradigm is often implemented through public relative performance feedback (PRF) [4], which entails disclosing workers’ productivity metrics in order to promote the diffusion of the practices adopted by top performers. Adopted by a large number of US corporations [5], PRF has measurably led to improvements in productivity in a variety of workplaces (e.g., hospitals [4, 6]), and has led to temporary improvements in test results when applied to students in schools [7].

On the other hand, experimental research suggests that PRF may lead to more nuanced outcomes under certain incentive schemes. Studies have shown that PRF may backfire in situations where participants are compensated under tournament-like incentives akin to schemes that are in place in many firms, where the top-performing employees receive a bonus [8]. Indeed, the cognitive costs associated with a change of strategy to adopt best practices often lead to further improvement for a minority of already excellent performers, and to a deterioration in performance for the rest of the population [9].

Similar contradictions also arise at the aggregate level of organisations. For example, the academic performance of higher education institutions is now measured and ranked in a variety of ways based, e.g., on the ability to attract funding, student output, awards received, graduate employment, etc. [10]. Although all these indicators individually contribute to the quality and prestige of academic institutions, their aggregation into rankings has attracted considerable controversy and criticism as a driver of homogenisation in higher education, as universities become more responsive to changes in the rankings themselves than to their broader social responsibilities [11, 12].

In line with the above considerations, a growing body of literature suggests that the outcomes of ranking processes do not necessarily reflect the true value of the individuals or organisations being ranked [13]. Arguably, such a disconnect between value and ranking is the byproduct of the interaction between three main factors: imitation, serendipity, and reactivity.

As mentioned above in relation to PRF, the imitation of ‘best practices’ adopted by successful individuals can backfire and exacerbate inequalities in performance. In fact, the disconnect between value and success is a typical emergent property of collective decision systems, where individual decisions are not made independently [14, 15]. Experimental studies have indeed demonstrated that the very same people or items can achieve markedly different levels of success in a ranking in situations where individuals can observe and imitate the choices made by others (see, e.g., [16] for a seminal example in an artificial cultural market). In such situations, the compound imitation of choices typically results in a very skewed visibility distribution, which in turn leads to a few dominant ‘hits’ ultimately capturing most of the attention, as is systematically the case for, e.g., movies [17], web pages [18], and even scientific papers [19].

At the same time, serendipity is known to play an exceedingly important - yet often underplayed - role in determining

success. **Serendipity refers to positive developments of events that occur in an unplanned manner. Notable examples of serendipity are scientific discoveries that have occurred in fortuitous ways, such as those of penicillin and X-rays.** A number of studies have highlighted how random events can lead to the aforementioned disconnect between an individual's value (e.g., skills and intelligence) and her level of success. For example, a recent simulation-based study has shown how short-term success in an artificial society is most often achieved by the luckiest individuals rather than the most talented ones [20], with real-world examples of similar dynamics having been found, e.g., in financial markets [21, 22], sports [23] and science [24].

In most cases, the tension between imitation and serendipity as different mechanisms to achieve success in a ranking is driven by reactivity, which refers to changes in behaviour due to the awareness of being observed (also referred to as the Hawthorne effect [25]). In this respect, quantifying and ranking performance is a self-defeating process in situations where individuals can partially manipulate the metrics according to which they are being ranked. This is encapsulated by the adage known as Goodhart's Law [26]: 'when a measure becomes a target, it ceases to be a good measure'. Examples of Goodhart's Law in action abound in many contexts. For example, surgeons in the UK reportedly try to avoid the most complex surgeries due to the introduction of public league tables reporting success rates [27]. Similarly, school systems based on standardized testing are known to be plagued by 'teaching to the test' practices, i.e., teachers devoting disproportionate amounts of time and resources to subjects known to be frequently assessed in tests, preventing pupils from receiving a broader education [28]. In recent years, academia has also been affected by similar practices due to the constantly increasing emphasis being placed on citation-based bibliometric indicators to quantify the impact of published research and rank researchers accordingly. Indeed, plenty of evidence relates such practices to the empirically observed increase in self-citation rates [29, 30] and exchange of citations between coauthors [31].

In contexts where individuals or institutions are ranked, reactivity provides a strong incentive to imitate the actions of top performers. Yet, as mentioned above, this can easily backfire. So, what should those lagging behind in a ranking do to climb closer to the top?

In the following, I propose a stylised model to contrast imitation and serendipity as competing mechanism in an artificial society whose agents are aware of being ranked based on their performance, and try to climb the ranking by either imitating the past actions of better ranked agents, or by trying their luck with the adoption of new actions that are presented to them at random. Within this simplified setting, I will seek to determine whether the adoption of best practices from top performers can always outperform luck, as one would intuitively expect. The model gives rise to rather rich dynamic, which unveils a negative feedback loop between the likelihood of climbing a ranking and the attempt of doing so through the imitation of top performers. Indeed, I will show that imitation is a largely self-defeating endeavour, which in most cases is vastly outperformed by serendipity.

The paper is organised as follows. In Section II I outline the model and provide some qualitative intuition on its functioning and its main results. Section III is then devoted to outlining such results in detail, while Section IV concludes the paper with a discussion on its implications.

II. THE MODEL

Let us consider N agents who repeatedly select which action to play among M possibilities. Each action can in principle yield a payoff up to a value $\pi_j \in [0, 1]$ ($j = 1, \dots, M$), but the agents' ability to reap the benefits of a particular action varies according to a matrix $\alpha_{ij} \in [0, 1]$ ($i = 1, \dots, N; j = 1, \dots, M$) such that the payoff that agent i receives when adopting action j reads

$$P_{ij} = \alpha_{ij}\pi_j . \quad (1)$$

In the following, and throughout the rest of the paper, I will assume both the π_j 's and the α_{ij} 's to be independent random variables drawn from a uniform distribution over $[0, 1]$.

The payoffs π_j in the above definition capture the intrinsic potential profitability of the available actions, whereas the factors α_{ij} capture the idiosyncrasies associated with the agents' abilities to profit from them due to, e.g., different skill sets. For example, in an academic context the payoffs π_j would quantify the overall potential for impact of a scientific field j , while α_{ij} would quantify the ability of a specific researcher i to publish high-quality research in it.

Crucially, there can be situations such that $P_{ij} > P_{ik}$ and $\pi_j < \pi_k$, i.e., cases in which an agent i is better off playing an action that is associated with a lower potential payoff, but still yields a higher *individual* payoff to her (if $\alpha_{ij} \gg \alpha_{ik}$). In the same analogy used above, this would represent a researcher i whose individual potential is much better fulfilled in a less impactful field.

At the beginning of time ($t = 0$) each agent starts out by playing a randomly selected action, but at any later time step ($t = 1, \dots, T$) has the opportunity to change action depending on the payoff she has received in the latest round.

Let us denote as $P_{ij}^{(t)}$ the payoff agent i has received by playing action j at time t . Let us define the payoffs an agent has accumulated over time as the agent's *utility*. This reads

$$u_i(t) = \sum_{t'=0}^t P_{ij}^{(t')} = \sum_{t'=0}^t \alpha_{ij,t'} \pi_{j,t'} , \quad (2)$$

and depends on the set of actions $\{\pi_{j_0}, \pi_{j_1}, \dots, \pi_{j_t}\}$ she has played at each round.

At each time step t , the agents are ranked based on the utility they have accumulated up to that point (i.e., $u_{i_1}(t) \geq u_{i_2}(t) \geq \dots \geq u_{i_N}(t)$), and use such ranking in order to decide whether to change their current action or not. Namely:

- (1) at each time step t each agent keeps playing the same action with probability equal to the last payoff she has received, i.e., $P_{ij}^{(t-1)}$;
- (2) if an agent changes action, then with probability $q \in [0, 1]$ she copies the time $t-1$ action of a randomly selected agent among those ranked better than her, while with probability $1-q$ she picks a new action at random.

The model's dynamic is sketched in Fig. 1. Point (1) above captures the agents' quest for actions that are profitable *to themselves*. Indeed, an agent will drop a potentially highly profitable action ($\pi_j \lesssim 1$) with high probability when her ability to benefit from it is low ($\alpha_{ij} \ll 1$). Point (2) instead describes the probabilistic selection process with which the agents choose new actions. This depends on one parameter $q \in [0, 1]$, which quantifies to what extent the agents pay attention to the ranking and choose to imitate the actions of their most successful peers. When q is large, the majority of action changes will be aimed at imitating the actions of the most successful agents. Conversely, when q is small most agents will select a random action when switching to a new one.

FIG. 1: Sketch of the model. At the beginning of a time step, agent i is playing action j , which awards a payoff π_j . With probability equal to the *individual* payoff $P_{ij} = \alpha_{ij}\pi_j$ (see Eq. (1)), the agent keeps playing the same action, and with probability $1 - P_{ij}$ switches to a new action, which is determined either via imitation or via serendipity. With probability q , the agent adopts the action being played by a (randomly selected) better ranked agent, while with probability $1 - q$ the agent selects a new action at random.

Fig. 2 provides some initial intuition of the results we can expect from the model. The panels illustrate agent trajectories in the space of possible actions for simulations of the model with $N = 200$, $M = 1000$, $T = 500$. The payoffs π_j and the matrix elements α_{ij} are independent and identically distributed variables drawn from a uniform distribution over $[0, 1]$. From left to right, the panels correspond to $q = 0.1$ (a model where the agents' action selection

process is largely random), $q = 0.5$, and $q = 0.9$ (a model where the agents' selection process is largely driven by the imitation of better ranked agents), respectively. The top (bottom) panels show the trajectories of the top (bottom) 10 agents according to the final ranking at time T .

FIG. 2: From left to right panels show results of a single simulation run of the model for $q = 0.1$, $q = 0.5$, and $q = 0.9$, respectively. Top (bottom) panels show the trajectories of the top (bottom) 10 agents based on the ranking at the final time $T = 500$. The y axis shows the numbers associated with the actions being adopted, which go from 1 to $M = 1000$ in no particular order. As it can be seen, in the $q = 0.5$ simulation the top agents tend to lock in on action 186, while in the $q = 0.9$ top agents lock in on action 516.

As it can be seen from the top panels, higher values of q lead to much higher stability in terms of the actions played by the highest ranked agents. Indeed, for $q = 0.1$ no particular pattern is clearly discernible, as the agents keep switching actions and do so mostly at random. On the other hand, for $q = 0.9$ the top 10 agents quickly lock in on the *same* action, and essentially keep playing it with very few interruptions. Furthermore, the ranking makes sure that such interruptions are short-lived, as top ranked agents only have a handful of peers to look up to in the ranking, and these are all playing the same action most of the time.

The bottom panels show that the above effect trickles down all the way to the bottom of the ranking. Indeed, it can be seen that for higher values of q even the lowest ranked agents tend to return to the action played by the highest ranked ones, albeit in a much more noisy fashion. All in all, these examples begin to highlight the presence of a clear feedback mechanism between the ranking and the level of attention the agents pay to it when switching actions: the higher the value of q , the more frequently all agents will turn to the ranking to make their decisions, *regardless* of their abilities. This mechanism dramatically narrows the diversity of choices made by top ranked agents, which in turn further narrows the options of lower ranked agents when they turn to the ranking to decide which actions to adopt.

Before proceeding to detail the model's results, it is important to acknowledge its main limitations. As outlined above, the agents' decision-making is based on two very simple rules, which mimic real-world behaviour on an intuitive level but lack microfoundations. In addition, the model does not allow for adaptive behaviour, in that the agents lack memory and therefore cannot learn from their past actions, and the actions' payoffs remain constant, regardless of the number of agents playing them and their position in the ranking. Therefore, the model is to be interpreted as a stylised representation of much more complex dynamics. Nevertheless, as we will see in the following sections, the model's strength precisely lies in the simplicity of its assumptions, which allow to draw meaningful comparisons between the model's results and real-world outcomes.

III. RESULTS

I will now turn to exploring the model's results in greater detail. First, I will investigate how utility is generated and distributed across the agents.

A. Utility and inequality

At any given time step we can define the total utility of the agents simply as $U(t) = \sum_{i=1}^N u_i(t)$, where the utility of each agent is defined as per Eq. (2). The left panel in Fig. 3 shows the total utility $U(T)$ at the end of simulations with $N = 200$, $M = 1000$, $T = 500$. As it can be seen, the total utility increases monotonically with the parameter q up to $q \sim 0.9$, after which it declines slightly. At first glance, this would seem to suggest that the agents are overall better off when changing actions based on the imitation of better ranked individuals. Yet, the increase in total utility is not unequivocally positive.

FIG. 3: Left panel: total utility $U(T)$ as a function of the parameter q . Central panel: Gini coefficient (see Eq. (3)) of the agents' utility distribution as a function of q . Right panel: utility accumulated by the bottom (blue circles) and top (purple diamonds) 10% of the agents in the ranking. In all panels circles / diamonds represent average values, while error bars represent 95% confidence level intervals obtained over 500 independent simulations. In all cases, simulations were run with $N = 200$ and $M = 1000$, and all values were measured at time $T = 500$.

The central panel in Fig. 3 shows the Gini coefficient for the distribution of the agents' individual utility at the end of simulations. A very well known measure of inequality in a society, the Gini coefficient is usually defined as

$$g(U(t)) = \frac{1}{NU(t)} \sum_{i < k} |u_i(t) - u_k(t)|. \quad (3)$$

By construction, the Gini coefficient ranges from 0 for a perfectly equal society ($u_i(t) = u_k(t)$, $\forall i, k$) to 1 for a completely unequal society where the entirety of the available utility is owned by a single agent ($u_i(t) = U(t)$ and $u_k(t) = 0$, $\forall k \neq i$). As shown in the Figure, the Gini coefficient $g(u(T))$ at the end of simulations increases monotonically with the parameter q , highlighting a steady increase in inequality.

The two above results together show that an increased attention towards the ranking drives towards an overall increase in utility, although such utility gets increasingly concentrated in the hands of fewer agents. In principle, this does not rule out that those at the bottom of the ranking might still be better off in absolute terms, in which case higher inequality would be a more justifiable outcome. However, the right panel in Fig. 3 shows that the total utility accumulated by the bottom 10% of the population eventually decreases for high values of q . Symmetrically, the total utility accumulated by the top 10% steadily increases with q .

In summary, the results presented in this section show that when imitation becomes the prevalent strategy, society as a whole becomes 'richer'. Yet, this is entirely driven by a much faster accumulation of utility in the higher layers of the ranking, whereas those at the bottom eventually accumulate less utility than they would if their actions were chosen at random.

B. Meritocracy and homogenisation

The results of the previous section show that when the agents pay more attention to the ranking, the inequalities between them increase. Yet, such an outcome would surely look more acceptable if it somehow reflected an underlying meritocratic dynamic, according to which the ‘best’ agents are those who accumulate more utility. In order to verify whether this is indeed the case, I introduce two measures of the agents’ intrinsic potential ability - which I refer to as *fitness* - based on the utility they would be able to extract if they selected actions based on two predetermined strategies. The first one is the average payoff an agent would receive if playing all actions uniformly at random

$$\phi_i^{\text{avg}} = \frac{1}{M} \sum_{j=1}^M P_{ij} = \frac{1}{M} \sum_{j=1}^M \alpha_{ij} \pi_j . \quad (4)$$

It should be noted that the above does *not* correspond to the average payoff an agent receives when $q = 0$, as even in that case agents preferentially play profitable actions and therefore do not sample the action space uniformly. The second fitness measure I consider is instead the highest payoff an agent can extract by playing her most profitable action, i.e.,

$$\phi_i^{\text{max}} = P_{ij^*} , \quad (5)$$

where j^* is such that $P_{ij^*} > P_{ij}, \forall j \neq j^*$.

The two above measures capture different aspects. The former considers agents with a more diversified portfolio of skills as the fittest, whereas the latter singles out those agents who excel at one specific action, regardless of their skills when playing other actions.

The left panel in Fig. 4 shows the Kendall rank correlation coefficient [32] between the ranking of the agents in terms of utility and the two above measures of fitness. The Kendall coefficient is defined in the range $[-1, 1]$, and quantifies the similarity between two ranked lists of objects by measuring the fraction of concordant pairs in the two lists (a pair is said to be concordant whenever, e.g., $\phi_i^{\text{avg}} > \phi_k^{\text{avg}}$, and $u_i > u_k$). Two observations can be made. First, unless q is very low utility systematically tends to correlate more with ϕ^{max} than with ϕ^{avg} , i.e., as soon as the ranking is relied upon to inform the agents’ decisions, then the agents who are rewarded the most tend to be those excelling in a single action (typically one associated with some of the highest payoffs) rather than those who are consistently good at playing several actions.

FIG. 4: Left panel: Kendall correlation between the agents’ total utility and their fitness, as defined in Eq. (4) (blue circles) and Eq. (5) (purple diamonds), as a function the parameter q . Right panel: society’s homogenisation, defined as the fraction $\delta(T)$ of the actions being played by at least one agent at the end of a simulation as a function of q . In both panels circles / diamonds represent average values, while error bars represent 95% confidence level intervals obtained over 500 independent simulations with $N = 200$, $M = 1000$, and $T = 500$.

Second, both correlations decrease with higher values of q . The correlation between utility and ϕ^{avg} does so monotonically, whereas the correlation between utility and ϕ^{max} decreases after reaching a maximum value around

$q \approx 0.3$. Such a decrease signals that the more the agents pay attention to the ranking, the less such ranking reflects the actual agents' skills, substantially reducing meritocracy.

The dynamics induced by the ranking also have consequences on society's homogeneity terms of the number of actions played by the agents. Let us denote the fraction of actions being adopted across the agent population at any given time as $\delta(t) = M^{-1} \sum_{j=1}^M \mathbf{1}(\pi_j, t)$, where the indicator function is such that $\mathbf{1}(\pi_j, t) = 1$ if at least one agent is playing action j at time t and $\mathbf{1}(\pi_j, t) = 0$ otherwise. Clearly, we have $\delta(t) \in [1/M; \min(1, N/M)]$, where the lower bound corresponds to all agents playing the same action, while the upper bound is attained when each agent is playing a different action.

The right panel in Fig. 4 shows that homogeneity increases with q by reporting the average value of $\delta(T)$, i.e. the average fraction of actions being played at the end of a simulation. As it can be seen, when the ranking plays no role in the agents' decisions ($q = 0$), the agents already discard a very large fraction of the available actions. Intuitively, this naturally happens through the agents' random search for higher payoffs: once an agents randomly 'stumbles upon' a highly rewarding action, she will keep playing it over multiple consecutive rounds with high probability. In contrast, with higher q the agents will increasingly tend to imitate the choices of their better ranked peers, ultimately shrinking the space of adopted actions to its bare minimum when $q \rightarrow 1$.

C. Ranking dynamics

How stable are the rankings produced by the model? In this section I address this question by studying **changes in the agents' ranking position**. Namely, I consider the fraction $m_i(t)$ of agents occupying a lower position than a certain agent i in the ranking at a given time t , i.e.,

$$m_i(t) = \frac{1}{N-1} \sum_{j \neq i} \Theta(u_i(t) - u_j(t)), \quad (6)$$

where $\Theta(\cdot)$ denotes Heaviside's step function (i.e., $\Theta(x) = 1$ for $x > 0$, and $\Theta(x) = 0$ otherwise), and quantify agent i 's **change in ranking position** over a time interval as $\Delta m_i(t, t + \Delta t) = m_i(t + \Delta t) - m_i(t)$ [33].

The panels in Fig. 5 show the time evolution of the above quantity averaged over different simulations with $N = 200$ and $M = 1000$. In all four panels, changes in the **ranking position as defined in Eq. (6)** are computed over time lags of $\Delta t = 100$ time steps, and the different panels refer to snapshots taken at time steps $t = 100, 200, 300, 400$. The solid lines represent averages, whereas the shaded regions represent 90% confidence level intervals, and each panel shows the results obtained for $q = 0.1$ and $q = 0.9$.

As it can be seen, at the beginning of its time evolution the **ranking changes substantially**, regardless of the agents' preferences when switching actions. Indeed, the downward trend in the top-left panel of Fig. 5 shows that agents that happen to start at the top of the ranking typically lose ground during early stages of the model's dynamics, whereas agents initially at the bottom tend to climb up. However, as the dynamics continues the agents' preferences become increasingly important, leading to very different outcomes.

When random choices are prevalent ($q = 0.1$), the model still allows for considerable **changes in the ranking** in the long run. On average, the position of most agents in the ranking does not change dramatically, and agents at the top rapidly consolidate their position, but large fluctuations still take place in the central and bottom parts of the ranking, allowing agents with less utility to climb up. Conversely, when the agents mostly imitate better ranked agents ($q = 0.9$), the ranking essentially freezes.

The above result highlights once more the negative feedback between rankings and active efforts to climb them based on the imitation of actions adopted by those at the top. Once the model has produced some early 'winners', they will keep their position at the top of the ranking, and any effort made by lower-ranked agents to beat them through imitation will only backfire. The only mitigation to this outcome is serendipity (low q), i.e. a random search for more profitable actions, which prevents lower-ranked agents from becoming perpetual 'losers'.

As a final remark, it should be noted that all of the above results are robust with respect to changes in the model's parameter specifications. That is, the model's behaviour as a function of q is qualitatively unaffected by changes in the values of N and M .

IV. DISCUSSION

This paper puts forward a stylised framework to model the emergence of a divide between the fitness of an agent and her position in a ranking according to a measure of performance. This is done by simultaneously accounting

FIG. 5: **Change in ranking position** (defined as the change Δm_i of the quantity in Eq. (6) over a time interval) for agents of a society with $N = 200$ and $M = 1000$, averaged over 500 independent simulations. In all panels, **the change in ranking position** is computed for each agent individually over consecutive sets of $\Delta t = 100$ simulation steps, and plotted as a function of the agent's position in the ranking m_i at the beginning of the interval Δt . As indicated by their labels, the panels refer to quantities computed across simulations at times $t = 100, 200, 300, 400$. Solid lines refer to averages, whereas the shaded regions denote 90% confidence level intervals.

for three well documented mechanisms. First, the agents' attempts at climbing the ranking account for reactivity, i.e., the awareness of being observed and changing behaviour accordingly. Second, the imitation of 'best practices' and most successful strategies is encoded in the agents' imitation of the actions adopted by top-ranked peers. Third, serendipity partially determines the agents' chances when they search for more profitable actions.

The combination of the three above factors gives rise to a fairly rich phenomenology, which allows to study the tradeoff between imitation and serendipity as different strategies to climb a ranking. In a nutshell, such a tradeoff can be summarised as follows. Attempting to climb a ranking by imitating the actions of those at the top is a self-defeating strategy that further consolidates the early advantage of a few lucky - and not necessarily talented - winners. Attempts based on serendipity, i.e., on a random search for more profitable actions, have instead a mitigating effect on these outcomes.

A number of considerations can be made on the above. First, it is interesting to notice how the model highlights the existence of a negative feedback loop between the attempt to 'enforce' meritocracy by means of a ranking process and the actual possibility of achieving it. The model's dynamic is such that it always creates some lucky winners, i.e., those agents who happen to stumble on a profitable action early on in the model's time evolution and keep playing it over several rounds with high probability. As shown in the left panel of Fig. 4, this is a general feature of the model, as there is always a weakly positive correlation between the success achieved by the agents and their overall fitness (echoing the findings of [20]). Yet, when decisions are largely driven by the ranking, most agents will seek to imitate the actions of the early lucky winners, with the only result of widening inequality (see Fig. 3) and ultimately reducing their own chances of climbing the ranking (see Fig. 5).

In this respect, it is interesting to observe that serendipity plays the role of a double-edged sword in the model. On the one hand, it hampers meritocracy by endowing a lucky minority of agents with an early - yet permanent - competitive advantage. On the other hand, when the parameter q is low, it partially restores meritocracy by favouring **upward mobility in the ranking** and a higher correlation between fitness and ranking outcomes. Put differently, luck

will always generate some disconnect between intrinsic skills and measured performance, but attempts to overcome this by means of a ranking process will typically make things worse for those lagging behind.

From the perspective of society as a whole, the attention paid to rankings has a number of effects. As already mentioned, when the agents seek to climb the ranking through the imitation of others, they ultimately generate higher inequality, lower meritocracy and reduced **chances of climbing a ranking** (see [34] for similar findings in the context of financial wealth accumulation). Furthermore, the imitation mechanism drastically reduces the diversity of the actions played by the agents, resulting in an almost complete homogenisation of society (see the right panel in Fig. 4). Interestingly, this echoes evidence from financial markets, where analysts often prefer to imitate each other's forecasts rather than independently coming up with their own [35].

Finally, what lessons can be learned from the above model in the context of academic research and higher education? At the level of individual researchers, the model's results are in line with empirical evidence from publication data, which reveals that the first papers in a novel field - regardless of their content - often tend to attract citations at a higher rate than the papers following them [36]. In this respect, quoting [36], 'the scientist who wants to become famous is better off - by a wide margin - writing a modest paper in next year's hottest field than an outstanding paper in this year's'. Paraphrasing this in the context of the model, those who aim to become top-cited scientists in their field have much better chances of doing it by serendipitously pursuing their own research agendas rather than by imitating those of already well-established scientists.

At the broader level of institutions, instead, the model sheds some light on the causes of the increased homogenisation of the higher education landscape, which indeed has been often associated with the ever-increasing emphasis put on university rankings [11, 12]. As shown in Fig. 4 (right panel), the more the agents' decisions are driven by the imitation of their top-ranked peers, the less the space of possible actions is explored, and all agents end up playing just a handful of actions, regardless of their profitability.

In this respect, as a **data-rich and ranking-driven environment**, academia represents the ideal laboratory where to test the model's predictions in future work. Indeed, the analysis of citation data easily allows to quantify the similarity of research outputs [37] and to follow individual career trajectories across a discipline's research space (see, e.g., [38]). These basic ingredients would allow to compare the impact achieved by 'trend-followers' who actively seek to publish in mainstream fields as opposed to serendipitous researchers who mostly follow their interests. Similarly, aggregating such data would allow to quantify the performance of academic institutions in relation to their strategic behaviour.

In conclusion, this paper puts forward a framework to model the interplay between imitation and serendipity in situations where individuals or organisations are ranked (and aware of being ranked) based on some quantitative metric of performance. **As mentioned above, the model is a deliberately stylised representation of the real-world dynamics of such situations and clearly has a number of limitations, which extensions of the present work will seek to overcome. Most importantly, future extensions will allow the agents to retain some memory of their previous choices and to learn from them, in order to adapt their preferences for imitation or serendipity accordingly. Also, it should be noted that the model's dynamics would not lead to significant changes if studied on a network of interactions (as opposed to the well-mixed population case considered here), due to the fact that the imitation of actions would be able to diffuse throughout it regardless of its specific topology. In this respect, future extensions of the model should make it more consistent with the available evidence that network structures alone can determine ranking outcomes [39].**

Nevertheless, the model's strength lies precisely in the clarity and simplicity of the assumptions made, and in the fact that these are enough to generate rich dynamics which qualitatively resemble real-world observations. Hopefully the model presented in this paper will contribute to reflect on the importance that we collectively place on rankings, and on the unintended consequences they may have on our societies.

Ethics statement

Not relevant to the present work.

Data accessibility

As a simulation study, no data have been used in this manuscript.

Competing interests

I declare I have no competing interests.

Authors' contribution

Not relevant to the present work.

Funding statement

I acknowledge support from an EPSRC Early Career Fellowship in Digital Economy (Grant No. EP/N006062/1).

Acknowledgments

I am thankful to Aleksandra Aloric, Fabio Caccioli, Simone Righi, and Valeria Vercesi for their feedback on preliminary versions of this manuscript.

Code availability

The code to simulate the model described in the paper has been uploaded to GitHub: <https://doi.org/10.5281/zenodo.3345835>

-
- [1] Y. W. Ramírez and D. A. Nembhard, *Journal of intellectual capital* **5**, 602 (2004).
- [2] S. Aral, E. Brynjolfsson, and M. Van Alstyne, *Information Systems Research* **23**, 849 (2012).
- [3] A. Rosenblat, T. Kneese, et al. (2014).
- [4] H. Song, A. L. Tucker, K. L. Murrell, D. R. Vinson, et al., *Harvard Business School Research Paper Series* pp. 16–043 (2015).
- [5] R. Nordstrom, P. Lorenzi, and R. V. Hall, *Journal of Organizational Behavior Management* **11**, 101 (1991).
- [6] H. Song, A. L. Tucker, K. L. Murrell, and D. R. Vinson, *Management Science* **64**, 2628 (2017).
- [7] G. Azmat and N. Iriberry, *Journal of Public Economics* **94**, 435 (2010).
- [8] R. L. Hannan, R. Krishnan, and A. H. Newman, *The Accounting Review* **83**, 893 (2008).
- [9] W. G. Gjedrem, *Journal of Behavioral and Experimental Economics* **74**, 1 (2018).
- [10] J. C. Shin, R. K. Toutkoushian, and U. Teichler, *University rankings: Theoretical basis, methodology and impacts on global higher education*, vol. 3 (Springer Science & Business Media, 2011).
- [11] S. S. Amsler and C. Bolsmann, *British journal of sociology of education* **33**, 283 (2012).
- [12] I. Peña-López et al. (2015).
- [13] J. Denrell and C. Liu, *Proceedings of the National Academy of Sciences* **109**, 9331 (2012).
- [14] S. Sinha and R. K. Pan, *Econophysics and Sociophysics: Trends and Perspectives* (2006).
- [15] J. Lorenz, H. Rauhut, F. Schweitzer, and D. Helbing, *Proceedings of the National Academy of Sciences* **108**, 9020 (2011).
- [16] M. J. Salganik, P. S. Dodds, and D. J. Watts, *Science* **311**, 854 (2006).
- [17] S. Sinha and S. Raghavendra, *The European Physical Journal B-Condensed Matter and Complex Systems* **42**, 293 (2004).
- [18] L. A. Adamic and B. A. Huberman, *Science* **287**, 2115 (2000).
- [19] S. Redner, *The European Physical Journal B-Condensed Matter and Complex Systems* **4**, 131 (1998).
- [20] A. Pluchino, A. E. Biondo, and A. Rapisarda, *Advances in Complex Systems* **21**, 1850014 (2018).
- [21] G. E. Porter (2004).
- [22] A. E. Biondo, A. Pluchino, A. Rapisarda, and D. Helbing, *PloS one* **8**, e68344 (2013).
- [23] R. H. Barnsley and A. H. Thompson, *Canadian Journal of Behavioural Science/Revue canadienne des sciences du comportement* **20**, 167 (1988).
- [24] G. Ruocco, C. Daraio, V. Folli, and M. Leonetti, *Palgrave Communications* **3**, 17064 (2017).
- [25] R. McCarney, J. Warner, S. Iliffe, R. Van Haselen, M. Griffin, and P. Fisher, *BMC medical research methodology* **7**, 30 (2007).
- [26] D. Manheim and S. Garrabrant, *arXiv preprint arXiv:1803.04585* (2018).
- [27] O. A. Jarral, K. Baig, C. Pettengell, R. Uppal, D. P. Taggart, A. Darzi, S. Westaby, and T. Athanasiou, *Circulation: Cardiovascular Quality and Outcomes* pp. CIRCOUTCOMES–116 (2016).
- [28] J. L. Jennings and J. M. Bearak, *Educational Researcher* **43**, 381 (2014).
- [29] M. Seeber, M. Cattaneo, M. Meoli, and P. Malighetti, *Research Policy* (2017).
- [30] M. Fire and C. Guestrin, *GigaScience* **8**, giz053 (2019).
- [31] W. Li, T. Aste, F. Caccioli, and G. Livan, *EPJ Data Science* **8**, 20 (2019).
- [32] M. G. Kendall, *Biometrika* **30**, 81 (1938).
- [33] M. Bardoscia, G. De Luca, G. Livan, M. Marsili, and C. J. Tessone, *Journal of statistical physics* **151**, 440 (2013).
- [34] Y. Biondi and S. Righi, *Journal of Economic Interaction and Coordination* **14**, 93 (2019).
- [35] O. Guedj and J.-P. Bouchaud, *International Journal of Theoretical and Applied Finance* **8**, 933 (2005).
- [36] M. E. Newman, *EPL (Europhysics Letters)* **86**, 68001 (2009).
- [37] V. Ciotti, M. Bonaventura, V. Nicosia, P. Panzarasa, and V. Latora, *EPJ Data Science* **5**, 7 (2016).
- [38] F. Battiston, F. Musciotto, D. Wang, A.-L. Barabási, M. Szell, and R. Sinatra, *Nature Reviews Physics* **1**, 89 (2019).
- [39] F. Karimi, M. Génois, C. Wagner, P. Singer, and M. Strohmaier, *Scientific reports* **8**, 11077 (2018).